# Sentinel Lymph Node Mapping and Biopsy in Cats with Solid Malignancies: An Explorative Study

**DOI:** 10.3390/ani12223116

**Published:** 2022-11-11

**Authors:** Lavinia Elena Chiti, Elisa Maria Gariboldi, Damiano Stefanello, Donatella De Zani, Valeria Grieco, Mirja Christine Nolff

**Affiliations:** 1Clinic for Smal Animals Surgery, Vetsuisse Faculty, University of Zurich, CH-8057 Zurich, Switzerland; 2Department of Veterinary Medicine and Animal Sciences, Università degli Studi di Milano, 26900 Lodi, Italy

**Keywords:** cat, feline, cancer, sentinel lymph node, staging

## Abstract

**Simple Summary:**

The sentinel lymph node (SLN) is the first node in the basin that drains a solid tumor, and its histopathological examination after surgical removal is recommended in dogs and humans for correct tumor staging. In cats, however, the implementation of SLN biopsy (SLNB) has never been described. In this study, we included 12 cats presented with 14 solid malignancies that underwent surgical excision of a solid tumor and SLNB. The mapping technique used, location and number of SLN, correspondence with the regional lymph node (RLN), and histological status of the SLN were determined. The detection rate and complications of SLNB were also recorded. Near-infrared fluorescence lymphography (NIRF-L) was performed in 64.3% of tumors and lymphoscintigraphy in 35.7%. The detection rate was 100% for both techniques. The SLN did not correspond (fully or partially) to the RLN in 71.4% of cases, with multiple SLN being excised in 9/14 tumors. No complications related to SLNB were recorded. At histopathology, metastases were identified in 41.7% of cats, all with MCT. SLNB guided by NIRF-L or lymphoscintigraphy seems to be feasible and safe in cats with solid tumors and should be suggested for correct tumor staging in cats, especially with MCT.

**Abstract:**

There is increasing evidence on the utility of sentinel lymph node (SLN) biopsy (SLNB) for the staging of dogs with various malignancies; however, comparable information is missing in cats. This multi-institutional study aims at reporting the feasibility and detection rate of SLNB guided by lymphoscintigraphy and the blue dye or near-infrared fluorescent lymphography (NIRF-L) in cats with solid tumors. In total, 12 cats presented with 14 solid malignancies that underwent curative-intent surgical excision of the primary tumor and SLNB were retrospectively enrolled. The mapping technique used, location and number of SLN, correspondence with the regional lymph node (RLN), and histological status of the SLN were retrieved. The detection rate and complications of SLNB were also recorded. NIRF-L was performed in 64.3% of tumors and lymphoscintigraphy in 35.7%. The detection rate was 100% for both techniques. The SLN did not correspond (fully or partially) to the RLN in 71.4% of cases, with multiple SLN being excised in 9/14 tumors. No complications related to SLNB were recorded. At histopathology, metastases were identified in 41.7% of cats, all with mast cell tumors (MCT). SLNB guided by NIRF-L or lymphoscintigraphy is feasible and safe in cats with solid tumors and should be suggested for correct tumor staging in cats, especially with MCT.

## 1. Introduction

Identification of tumor spread to the lymph nodes is one of the mainstays of correct tumor staging in dogs and cats [1,2,3,4,5,6,7]. Indeed, since the World Health Organization’s (WHO) Tumor-Node-Metastases (TNM) staging system was readapted for tumors in companion animals, increasing attention has been drawn to indications for modalities to and impact of the detection of nodal metastases [8]. Traditionally, evaluation of the lymph nodes has been based on palpation and fine-needle aspiration of the superficial basins. However, palpation and lymph node size alone are poor predictors of nodal metastases, and cytological examination can yield a significant rate of false negative results [9,10,11,12]. Histopathological examination on the excised nodes has been suggested as the most reliable modality to identify occult nodal metastases, both in dogs and cats [9,10,11,12,13]. While for several years, recommendations for lymph node staging in dogs and cats with solid malignancies included the removal and histopathological examination of the anatomically closest lymph node—the so-called regional lymph node (RLN)—more recently, the extirpation of the sentinel lymph node (SLN) instead of the RLN has been advocated [1,7,14,15,16]. The SLN is defined as the first node within the lymphatic basin that drains a primary tumor, and it is expected to be the first site to harbor metastases; its status can thus theoretically predict the status of the whole lymphatic basin with the greatest accuracy [17]. Indeed, given the capability of tumors to stimulate the new formation of lymphatic networks, the SLN may differ from the anatomically closest RLN, and with the assessment of the latter, there is thus a risk of missing nodal metastases and potentially understaging the patient [5,7]. In support of such inference, unpredictable patterns of nodal metastases have been reported in up to 22.3–63% of dogs with various tumor types [1,5,7,18,19,20]. The incorporation of SLN extirpation, guided by various mapping techniques, has been described for accurate assessment and staging of the lymphatic basin in dogs with mast cell tumors (MCT), mammary tumors, and head and neck tumors [1,2,5,7,12,21]. The advantages of the incorporation of SLN mapping and biopsy have been largely reported in human medicine and include accurate tumor staging and reduced morbidity compared to radical nodal dissection [1,7]. Similarly, in tumor-bearing dogs, a benefit in terms of accuracy in staging has been documented, and morbidity of the procedure is reportedly low, although studies that compare the morbidity of RLN vs. SLN extirpation are currently not available in the veterinary literature [1,7,22,23,24]. Disadvantages of SLN extirpation include the need for dedicated technologies and the learning curve for the mapping procedure, although the incorporation of SLN biopsy seems to be justified by the well-documented impact of correct nodal staging on prognosis and treatment recommendations for various tumor types [3,25,26,27]. Comparable data on the feasibility and advantages of incorporation of SLN mapping in the feline species are lacking. A tendency to lymphatic spread and nodal metastases is reported for various feline solid tumors [11,28,29]. In a recent study, metastases to the lymph nodes have been documented in 58.8% of cats with MCT, underscoring the importance of nodal assessment also in this species [30]. However, there is still considerable inconsistency among studies on indications and methods for nodal staging in cats, and most studies do not even include information on the status of the lymphatic basin [11,28,29]. Furthermore, information on the feasibility and impact on the staging of SLN mapping and biopsy in cats is limited to a few experimental studies [31,32] and a single clinical case of MCT [33]. 

This study aims to retrospectively assess the feasibility and detection rate of SLN mapping and biopsy guided by radiopharmaceutical and methylene blue or near-infrared fluorescent lymphography (NIRF-L) with indocyanine green (ICG) in a cohort of cats presented for staging and surgical treatment of solid malignancies. As a secondary aim, we evaluated the correspondence between the clinically expected RLN and the SLN and the rate of nodal metastases to the SLN to assess the impact of the procedure on correct tumor staging. We hypothesized that similar to what has been reported in dogs, both mapping techniques would yield high detection rates also in cats, that unpredictable patterns of lymphatic drainage can occur in cats with solid tumors, and that implementation of SLN biopsy could thus lead to upstaging compared to traditional cytological sampling or excision of the RLN.

## 2. Materials and Methods

This project was conducted as a retrospective multi-institutional study. Clinical records of client-owned cats presented for staging and surgical excision of cytologically/histologically confirmed solid tumors to two teaching hospitals (the University of Zurich, Zurich, Switzerland, and the University of Milan, Milan, Italy) from November 2018 to April 2022 were reviewed. 

Cats were included in the study if they underwent:Curative-intent surgical excision of the primary tumor; all tumors’ presentation were included: first presentation, recurrence, and T0; single and multiple presentations.Concurrent SLN mapping and extirpation, i.e., sentinel lymph node biopsy (SLNB)

At the time of treatment, owners had to sign a written consent for tumor removal, SLN mapping, excision protocol, and data collection. Any medical intervention on live animals was performed in respect of the National legislation for animal welfare. All interventions were conducted based on medical decisions. Based on the experience in canine patients and the growing body of evidence regarding the importance of lymphadenectomy in canine patients, these recommendations included SLN mapping and lymphadenectomy from 2018 in both centers.

Patient signalment (breed, sex, age, and body weight) and tumor clinical characteristics (location, size or tumor or scar, ulceration, first presentation or recurrence, single or multiple presentation) were retrieved from the clinical records. Before admission to surgery, all cats underwent a complete preoperative oncological staging following the current recommendations for each tumor type [34,35,36,37,38]. Imaging modality used to screen for distant metastases, results of cytology of abdominal organs (if performed), the clinical status of RLN (enlarged vs. normal-sized), and results of cytology of RLN (if performed) were collected.

Given the lack of specific studies on the lymphatic anatomy in cats, the location of the clinically expected RLN was predicted based on the lymphosomes’ concept developed for the dog [39] and readapted for the cat (Figure 1) and integrated with Baum et al. [40]. Sentinel lymph node mapping and biopsy were guided either by ICG-based NIRF-L (University of Zurich, Zurich, Switzerland) or radiopharmaceutical and methylene blue (University of Milan, Milan, Italy). Both techniques were performed as previously described in dogs [1,14].

For NIRF-L, ICG was injected peritumorally (integumentary tumors) or intratumorally right before aseptic preparation of the cat, and a dedicated fluorescent imager (IC-Flow^TM^ [Diagnostic Green GmbH, Aschheim-Dornach, Germany] or Visionsense VS3 Iridium^TM^ [Visionsense Ltd., Petah Tikva, Israel]) was used intraoperatively to visualize the fluorescent lymphatic drainage pathway(s) to the SLN(s) and guide its/their dissection. After removal of the SLN(s), the surgical field(s) was checked with the fluorescent imager for residual fluorescence. Every fluorescent lymph node in the surgical field(s) was removed, and the signal was checked again ex vivo. For lymphoscintigraphy, Technetium-99 metastable (Tc-99 m) labeled nano-sized human albumin was injected peritumorally in four quadrants, and planar regional static images were acquired preoperatively with a gamma camera (Picker Prism 2000XP, Picker International, Highland Heights, OH, USA) until the first draining lymphocenter(s) was visualized. Methylene blue was injected in the same fashion just before aseptic preparation, and intraoperatively the identification and dissection of the SLN(s) was guided by a hand-held gamma probe (Crystal probe SG04; Crystal Photonic GmbH, Berlin, Germany) and blue coloration. Every radioactive (“hot”) and every blue node was excised. After removal of the first SLN(s), the surgical field was checked for residual radioactivity and further nodes having a radioactive count of at least 10% of the SLN [1,5]. The mapping procedure was deemed successful if it allowed the extirpation of at least one hot and/or blue or fluorescent node for each tumor. Removal of adjunctive nodes that were neither hot, blue, or fluorescent that occurred during dissection of the SLN was also reported. The detection rate was calculated as the percentage of successful procedures on the total of procedures performed.

The following data regarding SLNB were retrieved: mapping technique used, location of the sentinel lymphocenter(s), and the number of excised SLNs correspondence between SLN and RLN. Correspondence was defined as follows: Total correspondence: all excised SLN would have been predicted based on the lymphosomes’ concept.Partial correspondence: the mapping procedure guided the extirpation of the RLN and at least one more node at an unpredictable site.Non-correspondence: all extirpated SLN occurred at different locations than the RLN.

Curative intent excision of the primary tumor was performed concurrently with SLNB. To avoid cross-contamination of healthy tissues with neoplastic cells, lymphadenectomy was either performed before tumor excision or, when tumor excision was performed first, surgical instruments and gloves were changed between the procedures. Any complication related to the administration of radiopharmaceutical, methylene blue, or ICG was also recorded.

Histopathological reports were revised, and the following variables were retrieved: tumor type, tumor grade (if reported), state of excisional margins (infiltrated, non-infiltrated), and metastatic status of the SLN [3,41,42]. In the case of MCT, Weishaar classification for nodal metastases developed in dogs was applied, and HN2–HN3 lymph nodes were considered metastatic [3]. All the histopathological diagnoses were made either by a board-certified or by a nationally certified specialized pathologist at the time of treatment.

Descriptive statistics were used to synthesize the distribution of retrieved variables. For categorical variables, the frequency of each modality is reported as the % of the total cases. For continuous variables, normality was assessed with Q–Q plots; median value, range, first quartile (Q1), and third quartile (Q3) are reported for non-normally distributed variables, while mean and standard deviation are reported for normally distributed variables. Statistical analysis was performed with open-source software: R-Software vers R 4.2.1, packages rms and PASWR; www.r-project.org (accessed on 25 September 2022). The significance level was set at *p* ≤ 0.05.

## 3. Results

In total, 12 cats with 14 solid tumors satisfied the inclusion criteria and were therefore included in the study. Included tumor types were: MCT (*n* = 9), oral squamous cell carcinoma (SCC) (*n* = 2), oral fibrosarcoma (*n* = 1), adenocarcinoma of the mandibular salivary gland (*n* = 1), and adenocarcinoma of the ceruminous glands (*n* = 1). Two cats with MCT were also included in a previous study [35]. There were ten domestic shorthair cats (83.3%), one Siamese (8.3%), and one Ocicat (8.3%). Eight cats were spayed females (66.7%), three were neutered males (25.0%), and one was an intact male (8.3%). The median age was 12 years (range 1.5–18 years; Q1:10 years–Q3:13 years). Bodyweight was available in all but one cat, with a median value of 4 kg (range 2.4–9.4 kg; Q1: 2.4 kg–Q3: 5 kg). Primary tumors were located on the head and neck in 11/14 cases (78.6%), distant limbs in 2/14 cases (14.3%), and proximal limbs in 1 case (7.1%). Eleven tumors (78.6%) were first presentations, two were microscopically infiltrated scars (T0) from previously excised oral SCC and MCT, and one (8.3%) was a local recurrence of a cutaneous MCT. One cat presented contemporary multiple MCT (*n* = 3), while the other eleven cats had single tumors. In both cats presenting with scars, radicalization was performed concurrently with SLN biopsy.

Tumor maximal diameter was available in 10/14 cases, with a median value of 4 mm (range 3–50 mm; Q1: 3 mm–Q3: 8.75 mm). One MCT was ulcerated. Only in one case (salivary gland carcinoma), the regional lymph node was enlarged. 

The preoperative staging consisted of abdominal ultrasound and fine-needle aspirates of the spleen and liver in all seven cats with MCTs. In two of these cats, cytology revealed mast-cell infiltration of the spleen, and splenectomy was thus performed concurrently with tumor excision and SLN biopsy. In four cases, contrast-enhanced whole-body CT was performed. Cytology of the RLN was performed in the cat with salivary gland adenocarcinoma and lymphadenomegaly and the cat with oral fibrosarcoma; metastases were cytologically evident in the first case (although not confirmed at histopathology on the excised node) and absent in the latter. 

Sentinel lymph node biopsy was guided by NIRF-L in 9/14 tumors (64.3%) and by radiopharmaceutical and methylene blue in the remaining 5 (35.7%) tumors (Figure 2). At least one SLN was successfully identified and excised for every included tumor, leading to a detection rate of 100% for both techniques.

A total of 28 lymph nodes were excised. Of these, 11 were hot and blue, and 14 were fluorescent. Three non-sentinel lymph nodes (one parotid and two mandibular nodes) were removed in three cats that underwent NIRF-L due to anatomical contiguity with the actual SLN. These nodes had no fluorescent signal ex vivo. In all three cases, however, the non-sentinel lymph node belonged to a lymphocenter from which at least another SLN was successfully removed. A total of 25 SLN from 21 sentinel lymphocenters were excised, of which 11 SLN were removed from 9 lymphocenters under the guidance of radiopharmaceutical and methylene blue, whereas 14 SLN belonging to 12 lymphocenters were excised with fluorescent guidance. Nine tumors were drained by multiple SLNs belonging to one (*n* = 3) or multiple (*n* = 13) lymphocenters, while the remaining five tumors were drained by a single SLN. The SLN fully corresponded to the RLN in 4 tumors (28.6%), while in the remaining 10 tumors (71.4%), at least one SLN occurring at a different location from the RLN was excised thanks to the mapping procedure. More specifically, correspondence was partial in four tumors and absent in three tumors, while in three cases, it was not possible to predict the location of the RLN because the tumor was located on the medial plane or between two lymphosomes. No complication or local or systemic side effects related to the administration of the tracer (radiopharmaceutical, methylene blue, or ICG) were recorded.

At histopathological analysis, nine tumors were confirmed as MCT (of which seven out of nine were classified as Sabattini‘s low grade, in one case, the grade was not available, and one was T0); two were confirmed as oral squamous cell carcinoma (SCC); one as oral fibrosarcoma (grade I); one as adenocarcinoma of the salivary gland; and one as adenocarcinoma of the ceruminous glands. Excisional margins were non-infiltrated in ten (71.4%) cases, including the two scars, and infiltrated in four cases (28.6%) (one squamous cell carcinoma and 3 MCT low grade). Nodal metastases were histopathologically detected in 8/25 SLN excised from 5/12 cats (41.7%), all with an MCT. None of the three non-SLN that were accidentally removed had evidence of metastases at histopathology. If considering only cats affected by MCT, five out of nine cats (55.6%) had nodal metastases (four cases HN3 and one case HN2). Tumor clinical and pathological characteristics, SLN location, and histological data are reported in Table 1. In Figure 3, the histopathological slides of MCT of cat n12 and the corresponding HN1 lymph node are exemplified.

## 4. Discussion

In the study presented here, we demonstrated that SLNB is potentially feasible in cats with solid tumors. We report a detection rate of 100% for two different mapping techniques, namely radiopharmaceutical combined with blue dye and NIRF-L, suggesting the reliability of both techniques for accurate nodal staging in cats. 

Prompt identification of nodal metastases is a mainstay of surgical oncology, given its impact on staging and, therefore, therapeutic suggestions and reliable prognostication [3,26,27,37,43]. Several investigations have been conducted in the last decade on the assessment of the lymphatic basin in dogs with various spontaneous tumors. Conversely, nodal staging is often neglected in the management of tumor-bearing cats, as there is common sense that nodal metastases are rare for most tumor types affecting this species [44,45]. In the available body of literature, there is considerable inconsistency in indications and methods for nodal staging in cats, and most studies do not even include information on the status of the lymphatic basin. However, a tendency to lymphatic spread and nodal metastases is also reported for feline solid tumors. Nodal metastases have been reported in around 30% of cats with maxilla-facial tumors [11,28,29] and in 35% of female cats with mammary gland malignant tumors [46]. Likewise, in cats with cutaneous MCT, nodal metastases have been detected in 20% of cats with Sabattini’s high-grade cutaneous mast cell tumors [41], and an association with guarded prognosis has also been demonstrated [47], further emphasizing the importance of correct nodal staging also in the feline species. In the study population, nodal metastases were identified in 41.7% of cats, all presented with MCT, leading to an estimated prevalence of nodal metastases of 55.6% when considering only cats with MCT. This percentage is higher compared with previous data from Sabattini’s study and could be the effect of the implementation of SLNB instead of traditional RLN assessment [41]. Indeed, in the present study, the actual draining node would have been totally or partially missed in 71.4% of cats without the implementation of a mapping technique, meaning that with the traditional assessment of the RLN, there would have been potential to miss nodal metastases in a significant proportion of cases. This result is comparable with a non-correspondence of the SLN and RLN ranging between 22.3 and 63% in dogs with various solid tumors [1,5,6,7,12,14,21,48,49]. It could thus be argued that the higher rate of nodal metastases that we report is due to an underestimation of the actual rates of nodal metastases in previous studies that assessed the RLN instead of the SLN. Considering the retrospective nature of the study, however, it could not be excluded that our result is due to some sort of selection bias, given that admission to the mapping procedure was decided by the attending clinician at the time of treatment. Further prospective and randomized studies are warranted to quantify the magnitude of the impact of SLN excision instead of RLN excision on the staging of cats with solid tumors, especially with MCT. 

Radiopharmaceutical and blue dye are considered the gold standard for SLNB in humans and have shown promising results in dogs, with detection rates as high as 95% in humans with oral tumors and 85.5% in women with breast cancer [32,50,51]. The same technique has shown promising results also in dogs, with detection rates of 83–100 for canine MCT and tumors of the head and neck [5,7,16]. The technique has not been previously reported in cats with spontaneous tumors, but it was recently described in eight cats and used as an experimental model for SLNB in humans’ head and neck SCC [32]. In this latter study, the cat was chosen as a big animal model because multiple inguinal and cervical nodes had been previously described in an anatomical study [32,52], suggesting a more similar anatomy to humans compared to other animal models. Of the eight cats that were included in this study, an SLN was successfully identified in six cases, with rapid migration of the nano-colloid in six out of eight cats and retention for up to twenty-four hours after injection; lymphatic drainage was not identified in two cats injected at the base of the tongue [32]. In our study, lymphatic drainage was detected in all cats, leading to a higher detection rate compared to the above-mentioned experimental study, although no cat with tumors of the tongue was included. 

Near-infrared fluorescence is an emerging technique in veterinary medicine. It has been proposed both in dogs and humans as a more accessible alternative to radiopharmaceutical, given that it does not involve the storage and administration of radioactive substances. In women with breast cancer, equal or even better sensitivity has been reported with NIRF-L compared to radiopharmaceutical [53]. Similarly, the technique has been proven successful for the identification of SLN in dogs with oral tumors [15] and with MCT [14]. Previous evidence of the implementation of NIRF-L for SLNB in tumor-bearing cats is limited to a case report in which the technique successfully guided the extirpation of draining nodes in a cat with multiple MCT [33]. The results of our study are promising, considering that a fluorescent lymphatic drainage pathway was identified transcutaneous in all included cases and that at least one fluorescent node was excised in all patients. Our results need, however, to be corroborated in future studies by addressing the detection rate for each technique on a wider sample of cats, possibly with a single tumor type. 

In the present study, three non-sentinel lymph nodes were accidentally excised during the dissection of the SLN due to anatomical continuity. A similar complication was recently described in dogs with MCT undergoing SLNB guided by radiopharmaceutical and blue dye and is reasonably due to a learning curve of the surgeons [1]. Remarkably, none of the non-sentinel lymph nodes was metastatic at the histopathological examination, suggesting that extirpation of the SLN only is sufficient for correct nodal staging. Such an assumption, however, should be taken cautiously given the low numerosity of the sample population and need further validation on a larger cohort of animals.

Finally, the fact that no complications or side effects related to the mapping procedures were recorded suggests that in cats, as in humans and dogs, both radiopharmaceuticals combined with methylene blue and ICG are safe for peritumoral administration. 

Limitations of the present study are related to the retrospective nature, the low numerosity of the sample, and the inclusion of cats with various tumor types, which precluded the possibility of statistically quantifying the impact of SLNB on tumor staging. Furthermore, the lack of long-term follow-up did not allow for the evaluation of the impact of SLNB compared to traditional assessment of the RLN or no nodal staging on the oncological outcome. Such limitations should be addressed in future prospective studies enrolling a larger number of cats and preferably including a single tumor type.

## 5. Conclusions

In conclusion, the results of the present study suggest that SLNB guided either by radiopharmaceutical combined with blue dye or by NIRF-L is feasible and reliable for nodal staging of cats with various solid tumors. Given a partial to total non-correspondence between the clinically expected RLN and the SLN in 71.4% of cases and an overall metastatic rate of 41.7% and 55.6% when considering only cats with MCT, implementation of a mapping technique should be included in the management of tumor-bearing cats, especially those with MCT, for correct nodal staging.

## Figures and Tables

**Figure 1 animals-12-03116-f001:**
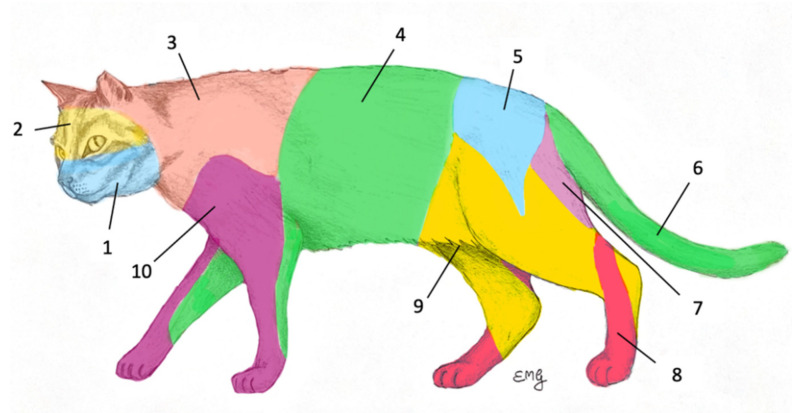
Readaptation of the lymphosomes’ concept developed by Suami et al. for the cat. 1, mandibular; 2, parotid; 3, dorsal superficial cervical; 4, axillary; 5, medial iliac; 6, lateral sacral; 7, hypogastric; 8, popliteal; 9, superficial inguinal; 10, ventral superficial cervical [39].

**Figure 2 animals-12-03116-f002:**
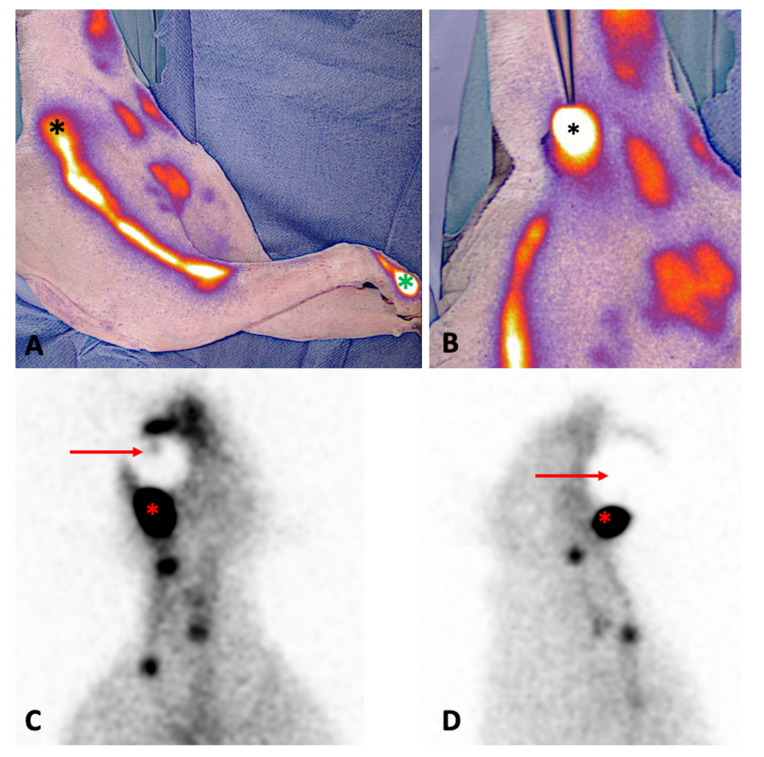
Sentinel lymph node mapping in cats. (**A**) Intraoperative fluorescent lymphography in cat n6: indocyanine green has been injected peritumorally in the third digit of the right hand (green asterisk), and the lymphatic pathway to the ipsilateral superficial cervical lymph node (black asterisk) is visible through the skin. (**B**) After skin incision, the superficial cervical node (black asterisk) is identified and dissected. (**C**) Preoperative planar lymphoscintigraphy in dorsoventral and (**D**) lateral static acquisitions in cat n10: the tumor is located at the left mandibula (red arrow); after peritumoral injection of radiopharmaceutical, the injection site is masked with a lead shield, and an uptake at the left mandibular lymph node (red asterisk) is observed.

**Figure 3 animals-12-03116-f003:**
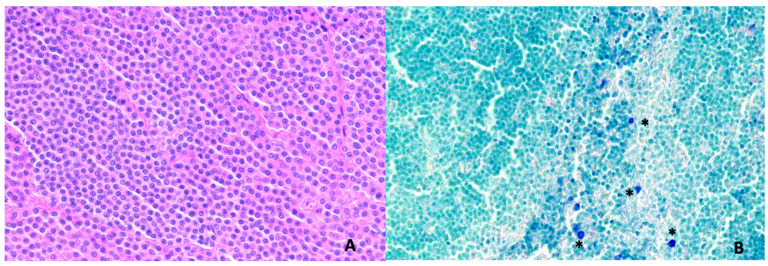
(**A**) Low-grade mast cell tumor in a cat (cat n12): round neoplastic cells, with moderate quantity of eosinophilic finely granular cytoplasm and vesicular round central/paracentral nucleus, are typically arranged in cords and sheets. Hematoxylin-Eosin stain 400×. (**B**) Sentinel lymph node in a cat with mast cell tumor (cat n12): the lymph node, graded as HN1 according to Weishaar et al. (2014), contains more than three individualized mast cells (asterisks) in sinuses and/or nodal parenchyma, in a minimum of four 400× microscopic fields. Giemsa stain 400×.

**Table 1 animals-12-03116-t001:** Tumor characteristics and regional and sentinel lymphocenter characteristics.

Cat Signalment	Tumor Location and Laterality	Tumor Histotype	Regional Lymphocenter	Mapping Technique	Sentinel Lymphocenters	Correspondence Sentinel and Regional Lymphocenter	SLN *n*	Histological SLN Status
Cat n.1 DS, SF, 10 years, 9.4 kg	Salivary gland, R	Salivary gland carcinoma	Retropharyngeal R	ICG	Retropharyngeal R *, Superficial cervical R	Partial	2	Neg
Cat n.2 DS, SF, 10 years, 5 kg	Ear, R	MCT, low grade	Superficial cervical R	ICG	Parotid R Superficial cervical R	Partial	3	HN3
Cat n.3 DS, IM, 9 years, 4.7 kg	Shoulder, L	Scar of excised MCT (grade not available)	Superficial cervical L vs. Axillary L	ICG	Superficial cervical L	Unpredictable	1	HN3
Cat n.4 DS, SF, 5 years	Tarsus, R	MCT, low grade	Popliteal R	ICG	Popliteal R	Correspondence	1	HN2
Cat n.5 DS, NM, 18 years, 3.5 kg	Ear canal, L	Ceruminous gland carcinoma	Retropharyngeal L	ICG	Retropharyngeal L	Correspondence	1	Neg
Cat n.6 DS, SF, 12 years, 2.4 kg	Hand third digit, R	MCT, low grade	Superficial cervical R	ICG	Axillary R	Non-correspondence	1	HN1
	Cheek, R	MCT, low grade	Mandibular R	ICG	Superficial cervical R	Non-correspondence	1	HN2
	Lip commissure, L	MCT, low grade	Mandibular L	ICG	Mandibular L Superficial cervical L	Partial	2	HN0
Cat n.7 DS, SF, 13 years, 2.4 kg	Eyelid, R	MCT, grade not available	Parotid R	ICG	Mandibular R	Non-correspondence	2	HN3
Cat n.8 DS, SF, 13 years, 5 kg	Chin, Median	SCC	Indeterminable R vs. L	Radio + MB	Mandibular L	Unpredictable	2	Neg
Cat n.9 DS, SF, 15 years, 3.8 kg	Ear, L	Fibrosarcoma	Parotid L	Radio + MB	Parotid L, Retropharyngeal L, Superficial cervical L	Partial	3	Neg
Cat n.10 DS, SF, 14 years, 4.8 kg	Mandibula, L	SCC	Mandibular L	Radio + MB	Mandibular L	Correspondence	2	Neg
Cat n.11 DS, SF, 13 years, 5 kg	Ventral to external ear, R	MCT, low grade	Parotid R and Superficial cervical R	Radio + MB	Parotid R Superficial cervical R	Correspondence	2	HN1
Cat n.12 DS, SF, 1.5 years, 4 kg	Chin, Median	MCT, low grade	Indeterminable R vs. L	Radio + MB	Mandibular R Mandibular L	Unpredictable	2	HN1

DS: Domestic shorthair cat; SF: spayed female; IF: intact female; NM: neutered male; IM: intact male; R: right; L: left; *: enlarged lymph node; MCT: mast cell tumor; SCC: squamous cell carcinoma; Radio: radiopharmaceutical technique; MB: methylene blue; ICG: indocyanine green; HN0: non-metastatic lymph node, HN1: pre-metastatic lymph node, HN2: early metastatic lymph node, and HN3: overt metastatic lymph node (based on Weishaar et al., 2014) [3].

## Data Availability

The data that support the findings of this study are available from the corresponding author upon reasonable request.

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
