# Peer review of "Sentinel Lymph Node Mapping and Biopsy in Cats with Solid Malignancies: An Explorative Study"

_animals, 2022, doi:10.3390/ani12223116_

Round 1

Reviewer 1 Report

The manuscript entitled "Sentinel lymph node mapping and biopsy in cats with solid 2 malignancies: an explorative study" sounds interesting and well-written. Limitations of the study are correctly stated. Discussion and conclusions appear to be interesting and useful to the audience of oncologists. Attached you will find the file with some corrections. Please check figure 1 at point 10, ventral superficial cervical. 

Author Response

Response to reviewer 1

The manuscript entitled "Sentinel lymph node mapping and biopsy in cats with solid 2 malignancies: an explorative study" sounds interesting and well-written. Limitations of the study are correctly stated. Discussion and conclusions appear to be interesting and useful to the audience of oncologists. Attached you will find the file with some corrections. Please check figure 1 at point 10, ventral superficial cervical. 

We thank the reviewer for the useful comments and suggestion. We have modified the manuscript accordingly.

Reviewer 2 Report

In this article, the author reported the feasibility and detection rate of SLNB guided by lymphoscintigraphy and the blue dye or near-infrared fluorescent lymphography (NIRF-L) in cats with solid tumors. Twelve cats presented with 14 solid malignancies that underwent curative-intent surgical excision of the primary tumor and SLNB were retrospectively enrolled. And found that SLNB guided by NIRF-L or lymphoscintigraphy is feasible and safe in cats with solid tumors and should be suggested for correct tumor staging in cats, especially with MCT. However, with editing and some major revisions, the manuscript's readability can be much improved to better convey the importance of your study.

1.      Line 37, what is the MCT? Line 44, what is the TNM? The author should give it a full name when it first appears.

2.      In the introduction section, the author did not systematically explain the necessity, advantages and disadvantages of the development of sentinel limp node. The author should make a supplementary explanation in the introduction section.

3.      Line 163, P value should be italicized in statistical analysis, please check the entire article for consistent formatting.

4.      In this manuscript, there is only one table to explain the author's views, and the evidence is insufficient. The author should add the corresponding real tumor images and pathological section images to verify the point of view.

Add: Pathological examination is the most basic method for the diagnosis of malignant tumors, even if the method of biopsy is adopted, the pathological lesions should be seen. Since the manuscript does the research of malignant tumor, the first diagnosis of malignant tumor is the first! Then we can talk about other issues. The author is requested to provide the most basic pathological diagnosis of malignant tumors.

5.      In this article, the grammar should be revised.

Author Response

In this article, the author reported the feasibility and detection rate of SLNB guided by lymphoscintigraphy and the blue dye or near-infrared fluorescent lymphography (NIRF-L) in cats with solid tumors. Twelve cats presented with 14 solid malignancies that underwent curative-intent surgical excision of the primary tumor and SLNB were retrospectively enrolled. And found that SLNB guided by NIRF-L or lymphoscintigraphy is feasible and safe in cats with solid tumors and should be suggested for correct tumor staging in cats, especially with MCT. However, with editing and some major revisions, the manuscript's readability can be much improved to better convey the importance of your study.

1. Line 37, what is the MCT? Line 44, what is the TNM? The author should give it a full name when it first appears.

Done

2. In the introduction section, the author did not systematically explain the necessity, advantages and disadvantages of the development of sentinel limp node. The author should make a supplementary explanation in the introduction section.

Done – a more detailed explanation on the utility of SLN biopsy has now been added in the introduction section

3. Line 163,P value should be italicized in statistical analysis, please check the entire article for consistent formatting.

 Done

4. In this manuscript, there is only one table to explain the author's views, and the evidence is insufficient. The author should add the corresponding real tumor images and pathological section images to verify the point of view.

We agree with the reviewer that histological diagnosis is important, and for this reason we have added an explanatory image of the pathological slide of a tumor and corresponding lymph node (now Figure 3). However, the request to include slides of all individual cases included in the study is unusual, and we are not aware of any current publication that ever did something comparable.  We could understand this request if we were publishing on a rare pathologic condition, but the main aspect of this work is instead to describe the imaging procedure (of which we decided to include more images – now Figure 2). The pathological diagnoses are additional information acquired from patient records, but they do not impact the finding of the study, which is that sentinel node mapping is feasible in cats. 

Furthermore, we do not understand which would be the added value of providing single slides of each tumor. How many slides does the reviewer want to see?  While one slide may suffice for histological diagnosis, it is not necessarily the case that more slides are needed to verify grading etc. It looks like the reviewer is asking for a second look pathological analysis: which impact would that on the findings of in vivo sentinel node mapping?

In addition, we would like to underline the fact that all the included pathological diagnoses were made by boarded or nationally certified specialized pathologists as part of the work up of the cats included in the study (explanation added at line 187-188). All cases included in the study prior to being sample population for a scientific publication are clinical cases that were handled aspiring to the highest scientific and clinical accuracy to provide the best available care and the best chance of optimal oncologic outcome. The clinical decision making was based on the pathological diagnosis at the time of treatment. The owners of these cats sign an informed consent and invest money to treat their cats. The diagnostic and therapeutic techniques included in this work are very expensive, and case selection for inclusion in these is very strict and draws on the most recent scientific publications

The request to view pathological slides is essential in the case of case report publications where the diagnosis of rare cases justifies their presence and embellishes the work. Requiring slides of individual cases on a common and simple histologic diagnosis such as feline mast cell tumor appears to be unusual, especially for a study in which establishing a pathological diagnosis is not within the aim of the study itself.

If the reviewer insists on this more than unusual request (as he or she questions the capability of our pathology laboratories), getting images of all cases would require contacting all involved pathology laboratories and request digitalization of all slides. If this is to be done, that will require additional author positions to compensate for the effort – plus at least 20 days of work. 

We do not believe that this is anyhow justified in a paper that is not focused on pathologic diagnosis, but on clinical imaging, and that is retrospective in nature – meaning that included cats have already been treated based on the declared diagnosis. 

Lastly, we want to emphasize, that the second reviewer made no such requests. 

We ask the editor to decide, if they really want to request over 20 images on a subject that is not the focus of the paper, considering that no other clinical study in the veterinary literature has so far provided such a conspicuous material.

5. In this article, the grammar should be revised.

Grammar has been revised.

Round 2

Reviewer 2 Report

The manuscript is improved. It can be accepted.